# The Effect of N, C, Cr, and Nb Content on Silicon Nitride Coatings for Joint Applications

**DOI:** 10.3390/ma13081896

**Published:** 2020-04-17

**Authors:** Luimar Correa Filho, Susann Schmidt, Cecilia Goyenola, Charlotte Skjöldebrand, Håkan Engqvist, Hans Högberg, Markus Tobler, Cecilia Persson

**Affiliations:** 1Division of Applied Materials Science, Department of Materials Science and Engineering, Uppsala University, 751 21 Uppsala, Sweden; luimar.filho@angstrom.uu.se (L.C.F.); charlotte.skjoldebrand@angstrom.uu.se (C.S.); hakan.engqvist@angstrom.uu.se (H.E.); 2Ionbond AG, Industriestrasse 9, Dulliken 4657, Switzerland; susann_Schmidt@gmx.de (S.S.); markus.tobler@ionbond.com (M.T.); 3Thin Film Physics Division, Department of Physics, Chemistry and Biology (IFM), Linköping University, 58183 Linköping, Sweden; cecigoyenola@hotmail.com (C.G.); hans.hogberg@liu.se (H.H.)

**Keywords:** silicon nitride, coating, joint replacement, wear, adhesion

## Abstract

Ceramic coatings deposited on orthopedic implants are an alternative to achieve and maintain high wear resistance of the metallic device, and simultaneously allow for a reduction in metal ion release. Silicon nitride based (SiN_x_) coatings deposited by high power impulse magnetron sputtering (HiPIMS) have shown potential for use in joint replacements, as a result of an improved chemical stability in combination with a good adhesion. This study investigated the effect of N, C, Cr, and Nb content on the tribocorrosive performance of 3.7 to 8.8 µm thick SiN_x_ coatings deposited by HiPIMS onto CoCrMo discs. The coating composition was assessed from X-ray photoelectron spectroscopy and the surface roughness by vertical scanning interferometry. Hardness and Young’s modulus were measured by nanoindentation and coating adhesion was investigated by scratch tests. Multidirectional wear tests against ultrahigh molecular weight polyethylene pins were performed for 2 million cycles in bovine serum solution (25%) at 37 °C, at an estimated contact pressure of 2.1 MPa. Coatings with a relatively low hardness tended to fail earlier in the wear test, due to chemical reactions and eventually dissolution, accelerated by the tribological contact. In fact, while no definite correlation could be observed between coating composition (N: 42.6–55.5 at %, C: 0–25.7 at %, Cr: 0 or 12.8 at %, and Nb: 0–24.5 at %) and wear performance, it was apparent that high-purity and/or -density coatings (i.e., low oxygen content and high nitrogen content) were desirable to prevent coating and/or counter surface wear or failure. Coatings deposited with a higher energy fulfilled the target profile in terms of low surface roughness (Ra < 20 nm), adequate adhesion (L_c2_ > 30 N), chemical stability over time in the tribocorrosive environment, as well as low polymer wear, presenting potential for a future application in joint bearings.

## 1. Introduction

Total joint replacements (TJRs) are surgical procedures carried out most frequently on patients suffering from arthritic pain or bone fractures [1,2,3,4,5]. These procedures are largely considered successful, with success rates up to 90% at 10 years follow-up for total hip replacements (THRs) and total knee replacements (TKRs) [6,7]. However, the aging and more active population places higher demands on these implants. 

Typically, for THRs the femoral head is replaced by a metal alloy (CoCrMo) or ceramic (zirconia-toughened alumina (ZTA)), and the acetabulum by a ceramic or polymer (ultrahigh molecular weight polyethylene (UHMWPE) or cross-linked polyethylene (XLPE)) [8,9,10,11,12]. TKRs are composed of a metallic femoral component (CoCrMo) as well as a polyethylene XLPE insert attached in the tray [3,13,14,15,16]. Ceramic coatings on metallic substrates can be used to reduce wear of structural materials, including manufacturing tools as well as joint implants [17,18]. Different types of ceramic coatings are under investigation for hip joint applications (e.g., TiN, DLC, ZrO_2_, ZrN, CrN, and Si_3_N_4_), while TiN and ZrN coatings are already in clinical use in knee implants [19,20,21,22,23]. These transition metal nitride coatings have the expressed purpose of extending the implant’s life time, by either preventing or minimizing the body’s immune reaction, which might result in osteolysis, aseptic loosening, and ultimately implant revision [24,25,26,27,28,29,30,31,32,33,34,35]. 

According to our previous work, SiN_x_ based coatings have shown potential as an alternative for joint bearings due to their biocompatibility, high wear resistance and hardness, and reduced metal ion release [36,37,38,39]. However, it is challenging to achieve an optimal combination of adhesion, coating density, and reactivity in SiN_x_ coatings; a high coating density resulting in a lower reactivity may give an insufficient adhesion to the substrate due to high residual stresses [40]. Alloying with a third element may be an option to improve the chemical stability while maintaining a balance in coating density and adhesion. For silicon nitride, previous studies have shown that the addition of Cr increases oxidation resistance and mechanical properties [41,42], while Nb improves the wear resistance and increases the hardness [43]. In previous studies, we reported that the addition of C altered the surface reactivity of silicon nitride and influenced the coating density and surface morphology [44,45]. In addition, we have shown that an increased N content results in a higher hardness and density [46,47,48]. In this study, we investigated the effect of N, C, Cr, and Nb content, as well as ion energy, on the properties of silicon nitride (SiN_x_)-based coatings for joint applications, with a focus on their wear performance in a hard-on-soft contact, since, as mentioned above, the counter surface in a joint implant is usually a polyethylene polymer. The coatings were deposited on top of Cr-based interlayers, and were evaluated in terms of chemical composition, surface roughness, mechanical properties, adhesion, and wear resistance in a hard-on-soft contact.

## 2. Materials and Methods 

### 2.1. Coating Deposition

Coating deposition was conducted in an industrial coating system, with a chamber volume of about 1 m³, equipped with four magnetrons, of which two were operated in unbalanced magnetron sputtering (UBM) and two in high power impulse magnetron sputtering (HiPIMS) mode. The coatings were deposited using 2-fold substrate rotation. The Si targets were operated at average powers of 5 kW and 8 kW in HiPIMS mode, while the Cr and Nb targets were operated in UBM mode at a sputtering power 1 kW for Cr and sputter powers of 1 kW, 2 kW, or 5 kW for Nb. SiN_x_ coatings with thicknesses ranging from 3.7 to 8.8 µm were deposited on mirror polished CoCrMo discs. Ion energies were controlled using three different bias voltages (low, medium, and high) as well as average target power settings. The sputter atmosphere was controlled at a pressure of 600 mPa, with N_2_-to-Ar ratios ranging between 17% and 40% and the remaining percentage to reach 100% was Ar. Detailed information can be found in Table 1. 

### 2.2. Compositional Analysis

The composition of the SiN_x_ coatings was investigated by X-ray photoelectron spectroscopy (XPS, Axis UltraDLD, Kratos Analytical, Manchester, UK) using monochromatic Al(K_α_) X-ray radiation (*hν* = 1486.6 eV). The base pressure in the analysis chamber during acquisition was < 1 × 10^−7^ Pa. The experimental conditions were such that the full width at half maximum (FWHM) of the Ag3d_5/2_ peak from the reference Ag sample was 0.45 eV. For all coatings, XPS survey spectra and core levels were recorded on as-received samples and after sputter cleaning. Sputter cleaning consisted of an initial step of 900 s at an Ar^+^ beam energy of 2 keV, followed by a second step for 900 s at an Ar^+^ beam energy of 4 keV. During sputter cleaning the Ar^+^ beam was rastered over an area of 3 × 3 mm^2^ at an incidence angle of 20°. Automatic charge compensation was applied throughout the acquisition, using low energy electrons provided by a flood gun. The composition of the coatings was assessed from XPS high-resolution core level spectra recorded from the Si 2p, Ar 2p, N 1s, C 1s, and O 1s regions after sputter cleaning. Core level spectra were analyzed with CasaXPS (v2.3.15, Casa Software Ltd, Teignmouth, UK). A Shirley-type background was subtracted, and the spectra were calibrated using adventitious surface carbon at 284.8 eV as a charge reference. For quantitative analysis of the metal-containing coatings the core levels of the Cr 2p and Nb 3d were applied for determination. The measurement precision for XPS analysis was ±5% for compositions below 10 at % and ±2–3% for compositions above 10 at % [49].

During wear tests and exposure to fetal bovine serum (FBS) solution a reaction occurred, and a white layer was formed on the surface of some coatings. This layer was examined using monochromatic Al (K_α_) X-ray photoelectron spectroscopy (XPS, Quantera II, Physical Electronics (PHI), Eden Prairie, MN, USA). Measurements were conducted on the surface after 2 min of sputtering Ar^+^ ions at 500 V and after an additional 20 min at 1 kV, to investigate the coating surface and further down in the coating, respectively. The sample was mounted on a glass slide in order to float the sample and automatic charge compensation was used throughout the measurement. Core level spectra were analyzed in CasaXPS, a Shirley-type background was subtracted, and the spectra were calibrated using adventitious surface carbon at 284.8 eV as a charge reference.

### 2.3. Surface Roughness

The coating roughness was measured before wear testing using optical profilometry, specifically vertical scanning interferometry (VSI) at 10× and a field of view (FOV) of 1.0. Each measurement corresponded to an area of 451 × 594 μm^2^. Typically, four measurements were performed on each sample to obtain R_a_ (arithmetic average). 

### 2.4. Nanoindentation

The hardness and elastic modulus of the coatings were measured in a CSIRO UMIS nanoindenter (Fischer-Cripps Laboratories, New South Wales, Australia) equipped with a three-sided Berkovich tip. All films were tested in the load-controlled mode and for calculations a Poisson’s ratio of 0.3 was used. For the tests, at least 30 indents with a load of 20 mN were performed [50]. 

### 2.5. Scratch Testing

In order to investigate coating adhesion, a scratch test was performed at different time points [51] using a scratch tester (CSEM-Revetest (CSEM, Neuchatel, Switzerland)) with a Rockwell C tip (apex 120°, tip radius 200 μm). A progressive load up to 100 N, at a loading rate of 120 N/min and a horizontal displacement rate of 6 mm/min were applied. This resulted in a scratch length of 5 mm, which was evaluated in a light optical microscope to determine the critical load L_C2_ indicating where the adhesion failure occurred [52,53]. Each sample was scratched three times at each time point. 

### 2.6. Wear Resistance (2D) in a Hard-on-Soft Contact

Multidirectional wear tests (MWT) were carried out to evaluate the response of the coatings against polyethylene using cylindrical pins with a nominal length of 19.1 mm and diameter of 9.5 mm. The pins were made of UHMWPE GUR1020 (one of the two most commonly used grades of UHMWPE in orthopaedics, defined as per BS ISO 5834-2 2019 [54]), provided by the collaborating industrial partners Peter Brehm GmbH (Weisendorf, Germany). MWT tests were performed in 0.2 μm filtered bovine serum solution (25%) at 37 °C. Prior to testing, the pins were presoaked in serum and cleaned according to standard [55]. The test was carried out with a nominal load of 150 N resulting in an estimated contact pressure of 2.1 MPa within the guidelines of [56], frequency 2 Hz, and sliding velocity 56 mm/s for 2,000,000 cycles (2.0 MC) using a 7 mm × 7 mm square path for a sliding distance of 28 mm/cycle. 

### 2.7. Statistical Analysis

IBM SPSS Statistics v 26 (New York, NY, USA) was used for all statistical analyses. An analysis of variance (ANOVA) was performed, followed by a Scheffe’s post hoc test. When Levene’s test for homogeneity of variances was significant, Welch’s robust test followed by a Tamhane post hoc test was used instead. The Pearson correlation coefficient was used to evaluate potential correlations. A critical level of α = 0.05 was used to determine significance.

## 3. Results and Discussion

### 3.1. Coating Thickness and Composition

The growth rate for the SiN_x_ coatings depended on applied target power settings and bias voltages, as well as the N_2_ and C_2_H_2_ gas flows. Increased SiN_x_ and SiMeN_x_ growth rates resulted from more material being removed from the target due to elevated target potentials [57]. Additionally, as Nb or Cr were added to the process, the number of operated targets increased and contributed to the SiMeN_x_ growth. The growth rate for the different bias settings showed a maximum at a medium level, indicating that the flux of film-forming species at low bias voltages was not optimally directed to the substrate table and resputtering occurred at high bias voltage settings. Increasing amounts of N_2_ led to decreased growth rates due to target poisoning [47] while an increased C_2_H_2_ gas flow resulted in an increased growth rate. This was attributed to a reduction in coating mass density and morphological density, specifically a pronounced growth of columns [45]. 

Following the trend of the coating growth rate, the coating composition changed by increasing N_2_ and C_2_H_2_ gas flows, leading to increased amounts of N and C in the coating. XPS results showed a nitrogen content close to 50 at % for all coatings (Table 2). SiN_x_ coatings with a higher ion energy and 40% of N_2_ during deposition yielded a nitrogen content in the coatings exceeding 51 at % and a N/Si ratio ≥1. This ratio has previously been shown to be beneficial to a lower dissolution rate, which could be advantageous to the coating lifespan [39]. When C_2_H_2_ was added to the process the coating showed higher O contents, which in turn supported the interpretation for the growth rate of SiCN_x_ at elevated C_2_H_2_ flows. Here, the reduction in coating density and the pronounced growth of columns led to incorporation of O as the coatings were exposed to air prior to analysis. Further, a reduced morphological density was observed as Cr and Nb were added to the process [40]. This was mirrored in higher O and C contents in the corresponding coatings.

The microstructure of similar coatings has been published earlier, for a range of coating parameters [47]. The SiN_x_ coatings that performed well were very dense and, thus, had low O and C contents, but also displayed more residual stresses. Likewise the coatings that did not perform well were less dense and contained more O and C (by adsorption) [48].

### 3.2. Surface Roughness

The average surface roughness (R_a_) determined for the as-deposited coatings was <50 nm (Table 3), thus fulfilling the standard for biomedical implants (ASTM F2033-12). Coatings *C-low*, *Standard*, *Nb-medium*, and *Nb-high* displayed the lowest values (7.69–12.97 nm), followed by coatings *Nb-low*, *Si Power-high*, *Bias-medium*, *C-high*, and i (14.71–19.97 nm). The highest values of R_a_ were obtained for coatings *Bias-high*, *N-medium*, *N-low*, and *N-high* (22.2–42.05 nm). As shown in Table 3, the coatings with higher Nb and C content presented relatively low surface roughness values. A lower surface roughness could possibly be attributed to the ionization energy of N being higher than C, which resulted in more carbon atoms being deposited. On the other hand more of the amorphous phase was being created, resulting in a smoother surface [58,59,60,61,62]. No statistically significant correlation could be found between surface roughness and coating thickness, nor between surface roughness and deposition rate.

### 3.3. Mechanical Properties (Nanoindentation)

The coating hardness varied from 13–25.4 GPa, with coatings *Standard*, *N-high,* and *N-low* exhibiting higher values (Figure 1). A similar tendency could be observed for the Young’s modulus.

Earlier studies on SiN_x_ coatings have determined similar values for hardness and Young’s modulus, although different deposition methods were applied [39,44,47,63,64]. A higher hardness suggests a higher coating density. Hardness values reported for other coatings for joint implants such as ZrN, TiNbN, Ox-Zr, and TiN coatings resided in a similar range, namely 14.0–31.0, 14.0–24.5 and 12.0–14.0 and 33–56 GPa, respectively [65]. 

### 3.4. Adhesion

The scratch test results in terms of Lc_2_ values are shown in Figure 2. As can be seen, coatings deposited with a higher target power showed lower L_C2_ values. This was due to higher residual stresses resulting from a higher N content and the increase in Si-N bonds [45,48]. Moreover, these coatings showed a generally denser morphology (data not shown), which in turn contributed to increased residual stresses [66], as demonstrated previously [48]. Furthermore, coatings with higher O and Cr contents displayed higher Lc_2_ values, which may be related to a lower amount of N-bonds, i.e., an opposite trend to that previously mentioned and/or a decreased coating density and, hence, residual stresses. The following coatings showed statistical differences: *Standard* vs. *C-high*, *Nb-medium*, *Nb-high*, *Bias-medium*, *Bias-high*, and *Si power high* as well as *C-high* vs. *Nb-medium*, *Nb-high*, *Bias-high*, and *Si power high*. 

### 3.5. Multidirectional Wear Tests

#### 3.5.1. Macroscopic Appearance and Surface Analysis

The macroscopic surface structure of the coatings after the wear tests is shown in Figure 3. The formation of an opaque layer on the surface could be observed during testing on some of the coatings (Figure 3). XPS measurements were, therefore, performed on coating *N-medium*, at a region that still displayed a reflective surface (assumed to be unworn) and a region that had formed an opaque layer on the surface. Previous work showed a tribofilm formation in aqueous environments for Si_3_N_4_ materials, and in those conditions a SiO_2_ and Si(OH)_2_ layer could be found, improving the wear resistance and reducing the coefficient of friction by acting as a self-lubricating layer [67,68,69]. However, in the XPS measurements herein the use of charge neutralizers and lack of a good charge reference made the positions of the peeks uncertain. To determine whether the Si2p and O1s peaks originated from Si-O bonds, the distance between the peaks, ΔE_b_, was determined and compared to the distance (ΔE_b_) from literature according to Briggs et al. [70]. The deconvoluted Si peaks were fitted with the smallest number of curves possible. The spectrum obtained at the surface revealed contributions attributed to Si-C (100.8 eV), Si-N (101.4 eV), and Si-O (102.8 eV), which correlated well with findings from similar materials. After 2 min sputtering at 500 V the Si-O contribution was no longer detected, while there were still contributions attributed to Si-N and Si-C, and after additional sputtering for 20 min at 1 kV only two contributions were identified, Si-N and Si-Si. These results indicated that the outer layer contained more O and C compared to the bulk of the coating, which could be due to the formation of a tribofilm during wear testing.

#### 3.5.2. Coefficient of Friction

Low coefficients of friction were observed for the first 0.5 million cycles for *N-high*, *Standard*, *Bias-medium*, and *Bias-high* (0.051–0.067). Coatings *N-low*, *Cr*, and *Si Power-high* showed somewhat higher values, from 0.103–0.108, with little variation. Coefficients of friction did not change markedly throughout the tests, except for coatings that reacted or failed during the test (Figure 4). This work generally showed lower coefficients of friction compared to previous work on Nb-Ti-N coatings (ranging from 0.11 to 0.12) and on TiN (0.14) [66]. 

#### 3.5.3. Volumetric Wear Rate

While *N-low* and *N-high* gave the lowest wear rates for the UHMWPE pins (< 0.37 mm^3^/MC, Figure 4), N-medium failed already in the first 0.5 MC (Figure 3), giving a high wear rate due to the increased surface roughness from the reacted surface. The *Standard* coating also gave a high wear rate, due to a reacted surface (Figure 3). The coatings with a higher C content all failed at 0.5 MC. Nb coatings failed at different time points, for example, *Nb-low* and *Nb-high* had failed at 2.0 MC and *Nb-medium* at 1.5 MC. The remaining coatings did not fail and presented low wear rates (0.74–3.63 mm^3^/MC). Figure 3 and Figure 5 show that coatings with no apparent reaction or coating failure, and that gave low pin wear rates, were those with an initially high hardness (22.5–28.4 GPa), and, hence, presumably higher density and lower reactivity and/or a high N content (*N-low*, *N-high*, *Bias-medium*, *Bias-high*, and *Si Power-high*), with the exception being the *Standard* (H = 23.4 GPa) coating, which, however, contained more oxygen than the well-performing coatings (Table 2), suggesting a higher reactivity. Coating wear through contact with UHMWPE during the tests was not expected, and coatings failing would rather be associated to a higher reactivity and subsequent dissolution [40]. 

## 4. Conclusions

Based on the results of the coatings tested in this work, some important conclusions were drawn. First, the low-ion energy coatings generally exhibited a lower hardness and initially higher critical load in scratch testing. High concentrations of impurities (higher O content and lower N content) were associated with early reactions and/or dissolution of the coating, as shown by XPS compositional analysis as well as multidirectional wear tests. During the wear tests coatings with lower or no apparent O content did not fail and showed a low volumetric wear rate of UHMWPE pins. SiN_x_ coatings of high N content, low O content (e.g., *N-high*, *Bias-medium*, *Bias-high*, and *Si Power-high*) are needed for the target–joint implant applications. Promising low wear rates were found for UHWMPE pins sliding against these latter coatings in a multidirectional wear test.

## 5. Patents

Ionbond AG, where S.S. and M.T. are employees, owns patents related to similar coatings. 

## Figures and Tables

**Figure 1 materials-13-01896-f001:**
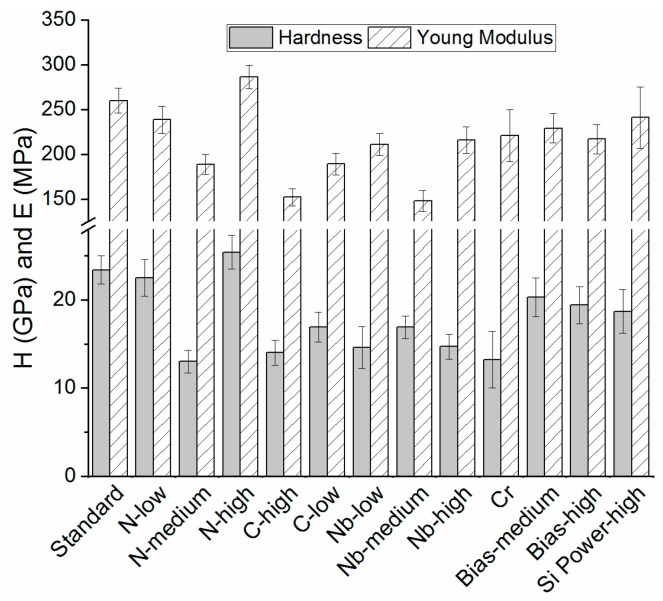
Hardness and Young’s modulus for SiN_x_ based coatings.

**Figure 2 materials-13-01896-f002:**
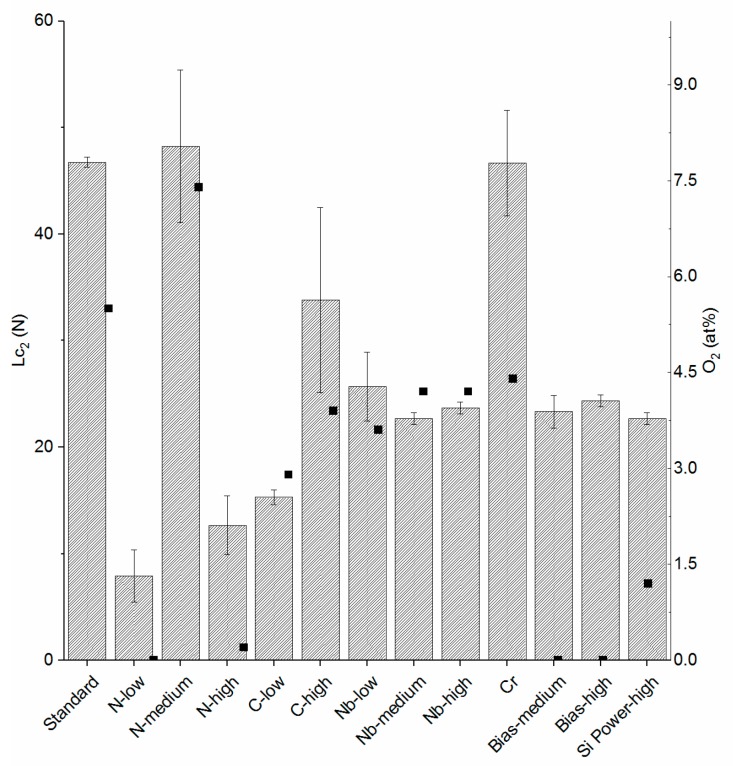
On the left axis the results for adhesion (Lc_2_) are shown for the coatings tested in this study, with bars and standard deviations. On the right axis the O_2_ content of the coatings is shown, represented by square dots.

**Figure 3 materials-13-01896-f003:**
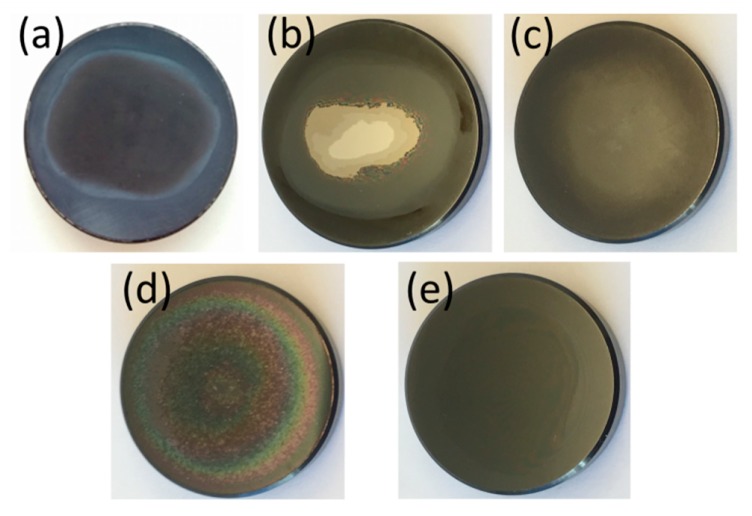
Typical macroscopic appearances of (**a**) a reacted surface (Standard), (**b**) a failed coating (Nb-medium), and (**c**,**d**) coatings with a surface layer: (**c**) Coating Cr and (**d**) coating Si Power-high. In (**e**) a Bias-high coating is shown, which did not present any layer formation or upcoming failure up to 2 MC.

**Figure 4 materials-13-01896-f004:**
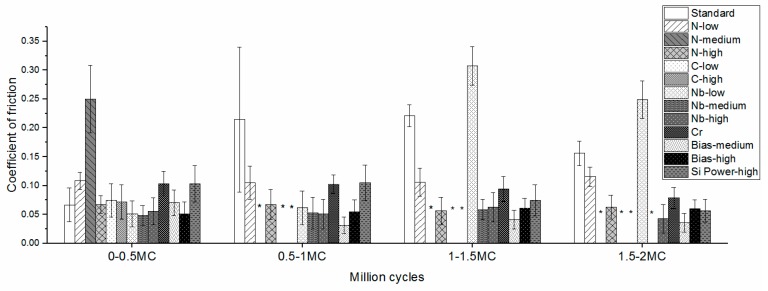
Coefficient of friction up to 2.0 MC for the tested coatings. During wear tests the following coatings wore through: (*) N-medium, C-high, and C-low at 0.5 MC; Nb-medium at 1.5 MC; Nb-low and Nb-high at 2.0 MC.

**Figure 5 materials-13-01896-f005:**
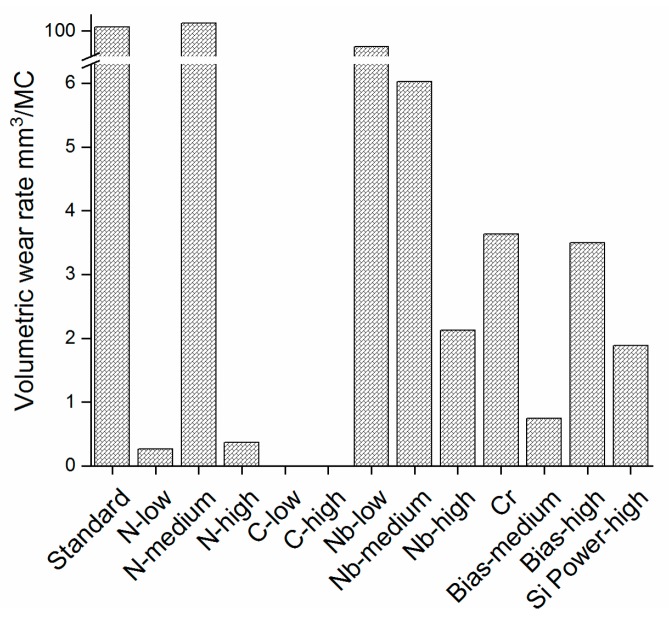
Volumetric wear rate for the UHMWPE pins ran against all coatings in this study. Coatings C-high and C-low showed negative values of −2.90 and −8.40 mm^3^/MC, respectively.

**Table 1 materials-13-01896-t001:** Description of the coatings and deposition processes used in this study. A pressure of 600 mPa was used for all deposition runs.

Analysis Aim	Coating Designation	Magnetrons				Bias	Gas	
		Si (HIPIMS)	Cr/Nb (UBM)	Si (HIPIMS)	Cr/Nb (UBM)		N_2_ Content	C_2_H_2_ Content
		(kW)	(kW)	(kW)	(kW)		(%)	(%)
Standard coating	Standard	5.0	-	5.0	-	low	40.0	-
Effect of N content in top layer	N-low	5.0	-	5.0	-	low	17.0	-
	N-medium	5.0	-	5.0	-	low	25.0	-
	N-high	5.0	-	5.0	-	low	40.0	-
Effect of C content	C-low	5.0	-	5.0	-	medium	38.0	2.5
	C-high	5.0	-	5.0	-	medium	36.0	4.0
Effect of Nb content	Nb-low	5.0	1.0	5.0	1.0	low	40.0	-
	Nb-medium	5.0	2.0	5.0	2.0	low	40.0	-
	Nb-high	5.0	5.0	5.0	5.0	low	40.0	-
Effect of Cr content	Cr	5.0	1.0	5.0	1.0	low	40.0	-
Effect of deposition energy	Bias-medium	5.0	-	5.0	-	medium	40.0	-
	Bias-high	5.0	-	5.0	-	high	40.0	-
	Si Power-high	8.0	-	8.0	-	low	40.0	-

**Table 2 materials-13-01896-t002:** Deposition settings, coating thicknesses, growth rates, and composition.

Coating Designation	Settings				Elemental Composition—XPS					
	Si Power	Bias	SiN/SiMeN/SiCN Thickness	Growth Rate	Si	N	O	C	Nb/Cr	N/Si
	(kW)		(µm)	(nm/s)	(at %)	(at %)	(at %)	(at %)	(at %)	-
Standard	5.0	low	5.5	0.13	42.8	49.7	5.5	2.0	-	1.16
N2-low	5.0	low	3.8	0.21	51.8	47.9	-	0.3	-	0.92
N2-medium	5.0	low	3.5	0.19	43.1	46.3	7.4	3.2	-	1.07
N2-high/Bias low/Standard	5.0	low	2.1	0.12	44.7	54.7	0.2	0.4	-	1.22
Bias-medium	5.0	medium	6.4	0.15	45.6	54.4	-	-	-	1.19
Bias-high	5.0	high	6.2	0.14	44.5	55.5	-	-	-	1.25
C-low, 2.5%	5	medium	5.4	0.17	33.1	47.2	2.9	16.9	-	1.43
C-high, 4%	5	medium	6.4	0.2	27.9	42.6	3.9	25.7	-	1.53
Nb-low, 2 × 1 kW Nb	5.0	low	3.1	0.17	39.9	46.8	3.6	1	7	1.17
Nb-medium, 2 × 2 kW Nb	5.0	low	3.1	0.17	33.9	45.2	4.2	1.3	13.7	1.33
Nb-high, 2 × 5 kW Nb	5.0	low	5.8	0.32	25.1	43.1	4.1	1.5	24.5	1.72
Cr-medium, 2 × 1 kW Cr	5.0	low	3.2	0.18	33.8	44.6	4.4	3.6	12.8	1.32
Si Power-high	8.0	low	4	0.24	46.5	52.3	1.2	-	-	1.12

**Table 3 materials-13-01896-t003:** Average surface roughness of SiNx coatings, as measured by interferometry. Coatings attributed with the same letters from a–e were not statistically significantly different (i.e., p > 0.05).

Coating Designation	R_a_ (nm)	Statistical Differences
Uncoated CoCr	3.5 ± 0.2	*a*
Standard	10 ± 0.9	*b, c*
N-low	33.2 ± 2.9	*-*
N-medium	33.1 ± 12.4	*e*
N-high	42.0 ± 6.0	*-*
C-low	7.6 ± 0.5	*a, b*
C-high	18.8 ± 3.0	*d, e*
Nb-low	14.7 ± 0.7	*c, d*
Nb-medium	12.9 ± 0.4	*c*
Nb-high	10.1 ± 3.4	*b, c*
Cr	19.9 ± 0.8	*e*
Bias-medium	18.0 ± 1.1	*d, e*
Bias-high	22.2 ± 1.1	*-*
Si Power-high	16.8 ± 0.5	*-*

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
