# Peer review of "The Effect of N, C, Cr, and Nb Content on Silicon Nitride Coatings for Joint Applications"

_materials, 2020, doi:10.3390/ma13081896_

Round 1

Reviewer 1 Report

The paper is worthy of presentation in this journal and has a large number of experimental observations that will prove useful for other scientists to replicate.  However, the manuscript can be improved using the following suggestions:

  1. The spacing between lines in the 'abstract' section seems too large, should be corrected;
  2. Table 1 is too big for the page, it should be reduced to fit the page;
  3. A 'discussion' section should be added to describe the results then followed by a 'conclusion' section.
  4. Reference spacing seems to be too large, reduce the spacing.

Also, check the grammar and spelling throughout by a native English speaker.

Author Response

Dear editors,

We are grateful for the attention paid to our manuscript by yourself and the reviewers. We have taken all comments from the reviewers into account and reproduce them below, together with our answers (in red). Changes to the manuscript have been marked in yellow in the text.

Reviewer 1:

The paper is worthy of presentation in this journal and has a large number of experimental observations that will prove useful for other scientists to replicate.  However, the manuscript can be improved using the following suggestions:

  1. The spacing between lines in the 'abstract' section seems too large, should be corrected;

Reply:

This has been corrected and we will follow the advice from the journal editors in terms of other layout considerations.

  1. Table 1 is too big for the page, it should be reduced to fit the page;

Reply:

This has also been corrected.

  1. A 'discussion' section should be added to describe the results then followed by a 'conclusion'

Reply:

We have chosen to keep the Results and Discussion section together in the paper, which is an option in this journal. The manuscript does contain a Conclusions section already.

  1. Reference spacing seems to be too large, reduce the spacing.

Reply:

The reference spacing has been reduced.

  1. Also, check the grammar and spelling throughout by a native English speaker.

Reply: The grammar and spelling has been checked and we will follow the advice by the editor regarding language.

Author Response

Dear editors,

We are grateful for the attention paid to our manuscript by yourself and the reviewers. We have taken all comments from the reviewers into account and reproduce them below, together with our answers (in red). Changes to the manuscript have been marked in yellow in the text.

Reviewer 2:

The authors made and tested SiNx coatings with varying dopants to determine which composition would have the best mechanical wear and corrosion resistance in a tribocorrosive environment. The authors provided a thorough explanation of their methods, including the statistics used. Give the large amount of data produced, the trends are often obscured in the graphs. The paper would be greatly improved with some minor adjustments to the figures and tables and to the introduction. Please see below.

The following edits are recommended:

  1. The introduction contains a lot of superfluous information about the typical material design for the total joint replacements that does not help inform the rest of the paper. Recommend using this space to help build the motivation for the tests done in the paper. For example, it is unclear why roughness is an important parameter.

Reply:

While we believe that the information on the typical material design for TJRs is important information to keep in order to put the developed materials in context, we agree with the reviewer that this information could be better organized and that the specific work of this paper could be more clearly motivated in the introduction. The introduction was therefore modified accordingly, to also clarify that (and why) a soft counter surface was used in the wear test (which is also why the coating roughness is specifically important).

  1. Reference 44 is for different coating system than SiNx. Recommend keeping the reference but changing the wording to clarify this point.

Reply:

We are sorry for this unclarity. We have removed this reference to avoid any confusion.

  1. Tables 1, 2, and 3 are hard to read. Please add borders along the rows and delineate the coatings that were designed to compare a specific component. For example, the N2-low, N2-medium, N2-high should be grouped together with color or some other border so the reader can compare them easily.

Reply:

We agree with the reviewer that the tables could be clarified and have added lines as suggested by the reviewer. However, the tables are part of the journal template and we have therefore refrained from further modifications.

  1. While the text provided good descriptions the coating behavior trends, the figures alone do not capture the important information needed to identify those trends. Can you change the graphs to highlight the trends? For example, in Figure 2 can you color the bars with a scale that indicates the oxygen content? That way the reader can see quickly the trends that you have identified without having to refer back to the table.

Reply:

We agree with the reviewer and Figure 2 was modified to include the oxygen content of the coatings.

Reviewer 3 Report

The manuscript presents a very interesting topic and with results that will certainly be very useful for the scientific community. However, I think the paper would be much more appealing if the authors included some microstructural characterization.

Author Response

Dear editors,

We are grateful for the attention paid to our manuscript by yourself and the reviewers. We have taken all comments from the reviewers into account and reproduce them below, together with our answers (in red). Changes to the manuscript have been marked in yellow in the text.

Reviewer 3:

The manuscript presents a very interesting topic and with results that will certainly be very useful for the scientific community. However, I think the paper would be much more appealing if the authors included some microstructural characterization.

Reply:

We agree that microstructural characterization of the coatings is of interest, and we have added references to published data on similar coatings in sections 3.1 and 3.4.

Round 2

Reviewer 1 Report

Acceptable after revisions